# Large-scale two-photon imaging revealed super-sparse population codes in the V1 superficial layer of awake monkeys

**Shiming Tang[1,2,3]\*, Yimeng Zhang[4,5], Zhihao Li[4,5], Ming Li[1,2,3], Fang Liu[1,2,3], Hongfei Jiang[1,2,3], Tai Sing Lee[4,5]\***

[1]School of Life Sciences and Peking-Tsinghua Center for Life Sciences, Peking University, Beijing, China; [2]IDG/McGovern Institute for Brain Research, Peking University, Beijing, China; [3]Key Laboratory of Machine Perception, Peking University, Beijing, China; [4]Center for the Neural Basis of Cognition, Carnegie Mellon University, Pittsburgh, United States; [5]School of Computer Science, Carnegie Mellon University, Pittsburgh, United States

**Abstract** One general principle of sensory information processing is that the brain must optimize efficiency by reducing the number of neurons that process the same information. The sparseness of the sensory representations in a population of neurons reflects the efficiency of the neural code. Here, we employ large-scale two-photon calcium imaging to examine the responses of a large population of neurons within the superficial layers of area V1 with single-cell resolution, while simultaneously presenting a large set of natural visual stimuli, to provide the first direct measure of the population sparseness in awake primates. The results show that only 0.5% of neurons respond strongly to any given natural image — indicating a ten-fold increase in the inferred sparseness over previous measurements. These population activities are nevertheless necessary and sufficient to discriminate visual stimuli with high accuracy, suggesting that the neural code in the primary visual cortex is both super-sparse and highly efficient.

DOI: https://doi.org/10.7554/eLife.33370.001

**\*For correspondence:**
tangshm@pku.edu.cn (ST);
tai@cnbc.cmu.edu (TSL)

**Competing interests:** The authors declare that no competing interests exist.

## Introduction

The efficient-coding hypothesis is an important organizing principle of any sensory system (*Barlow, 1981*; *Olshausen and Field, 1996*). It predicts that neuronal population responses should be sparse, although the optimal level of sparseness depends on many factors. Most of the experimental evidence of sparse coding comes from the responses of individual neurons that were exposed to a large set of natural image stimuli, measured using single-unit recording techniques (*Haider et al., 2010*; *Hromádka et al., 2008*; *Rust and DiCarlo, 2012*; *Vinje and Gallant, 2000*). The sparseness of a neuron's response to a large set of stimuli in these studies was used to infer the sparseness of the population responses. The first direct measurement of population response sparseness was performed with two-photon (2P) GCaMP6 signal imaging in rodents (*Froudarakis et al., 2014*). One potential confound in this work, however, is that GCaMP6 responses are subject to saturation at neuronal firing rates above 60–80 Hz (*Chen et al., 2013*; *Froudarakis et al., 2014*), leading to the potential to under-estimate the sparseness measures that can capture the peakedness or sharpness of the population response distributions. Thus, direct and accurate measurement of the population sparseness of neuronal response, particularly in non-human primates, is required.

In this study, we provided the first direct measurement of population sparseness from V1 of awake macaques. We performed 2P imaging on a large population of neurons using the genetically encoded calcium indicator GCaMP5 (*Akerboom et al., 2012*; *Denk et al., 1990*), delivered with

adeno-associated viruses (AAVs). We showed previously that GCaMP5 exhibits linear non-saturating responses across a wide range of firing rates (10–150 Hz) (*Li et al., 2017*), allowing us to measure accurately the response sparseness of almost all of the neurons in layer two in V1 within a 850 μm x 850 μm field of view—the spatial scale of about one hypercolumn.

## Results and discussion

Our 2P imaging of GCaMP5 recorded neuronal population calcium ($Ca^{2+}$) responses to 2250 natural images in V1 layer two in two awake macaques (about 1000 neurons each). Each monkey performed a fixation task while stimuli were presented to the appropriate retinotopic position in the visual field. Each trial sequence consisted of: a blank screen presented for one second, followed by a visual stimulus for one second. Each activated cell's region-of-interest (ROI) was defined as the compact region (>25 pixels) in which brightness exceeded three standard deviations (stds) above baseline, for each individual differential image. The standard ratio of fluorescence change ($\Delta F/F0$) of each of these regions of interest (ROIs) during stimulus presentation was calculated as the neuron's response (see 'Materials and methods').

The receptive fields (RFs) of the neurons were characterized using oriented gratings and bars presented in various positions. The RF centers of the imaged neurons were located between 3° and 5° in eccentricity. In each trial, a stimulus of 4° x 4° in size was presented, randomly drawn from a set of 2250 natural image stimuli (*Figure 1*, *Figure 1—figure supplement 1c*). The entire set of stimuli was repeated three times. These natural images evoked robust visual responses in the imaged neurons (*Figure 1a,b*, *Figure 1—figure supplement 1c*).

We recorded data from 1225 neurons in monkey A and 982 neurons in monkey B, and found that only a few neurons in each monkey responded strongly to any given image (*Figure 1a,b*), though most of the neurons responded strongly to at least one of the images in the set (*Figure 1c*). The rank-ordered distributions of the population responses were always sharply peaked (*Figure 1d,e*). On average, the percentage of neurons that responded above each's half-maximum was 0.49% (6.0/1,225) for monkey A and and 0.42% (4.1/982) for monkey B (*Figure 1f,g*). This is a measure of population sparseness that can capture the peakedness of the population response distribution (see 'Materials and methods'). In other words, only about 0.5% of the cells responded *strongly* to each image, indicating a very high degree of sparseness in the strong population responses or a very sharp population response distribution.

We also examined each neuron's stimulus specificity or life-time sparseness (*Willmore et al., 2011*). Interestingly, we found that most cells responded strongly to only a small number of images in the whole stimulus set (*Figure 2a,b*). The preferred images for individual neurons often shared common features. For example, neuron 653 of monkey A was most excited when its receptive field (0.8° in diameter) covered the lower rim of the cat's eye (indicated by the red dashed line in the inset in *Figure 2b*). The neuron's preference for the specific curve feature was further confirmed by checking its selectivity to a variety of artificial patterns (*Figure 2b*). Similarly, neuron 949 of Monkey A was selective to a different specific curvature embedded within the natural stimuli (*Figure 2d*). A more systematic characterization of these cells using artificial stimulus patterns has been reported previously (*Tang et al., 2018*), and this work showed that many of these neurons are highly selective to specific complex patterns. To measure the sharpness of each neuron's stimulus tuning curve — its life-time response sparseness — we computed the percentage of the stimuli that excited each neuron to >50% of the peak response found from the entire stimulus set. This population average was 0.49% (11/2,250) for Monkey A and 0.41% (9.3/2,250) for Monkey B (*Figure 2e,f*). This suggests that a high degree of stimulus specificity or life-time sparseness goes hand-in-hand with a heightened population sparseness.

To understand how much information was carried in the sparse ensemble of population activities, we evaluated how well the sparse neural responses allow a decoder to discriminate the 2250 stimuli (*Quian Quiroga and Panzeri, 2009*; *Froudarakis et al., 2014*). The entire population's decoding accuracy was 54% for Monkey A and 38% for Monkey B, whereas the chance accuracy was 0.04% (1/2,250) (*Figure 3a,b*, horizontal dashed lines). These population decoding accuracies estimate the *highest achievable decoding performance* from the activities recorded for the full population, and serve as the upper limit of accuracy performance that can be achieved by decoders that are made from any subset of neurons in the population (*Figure 3a,b*, horizontal dashed lines).

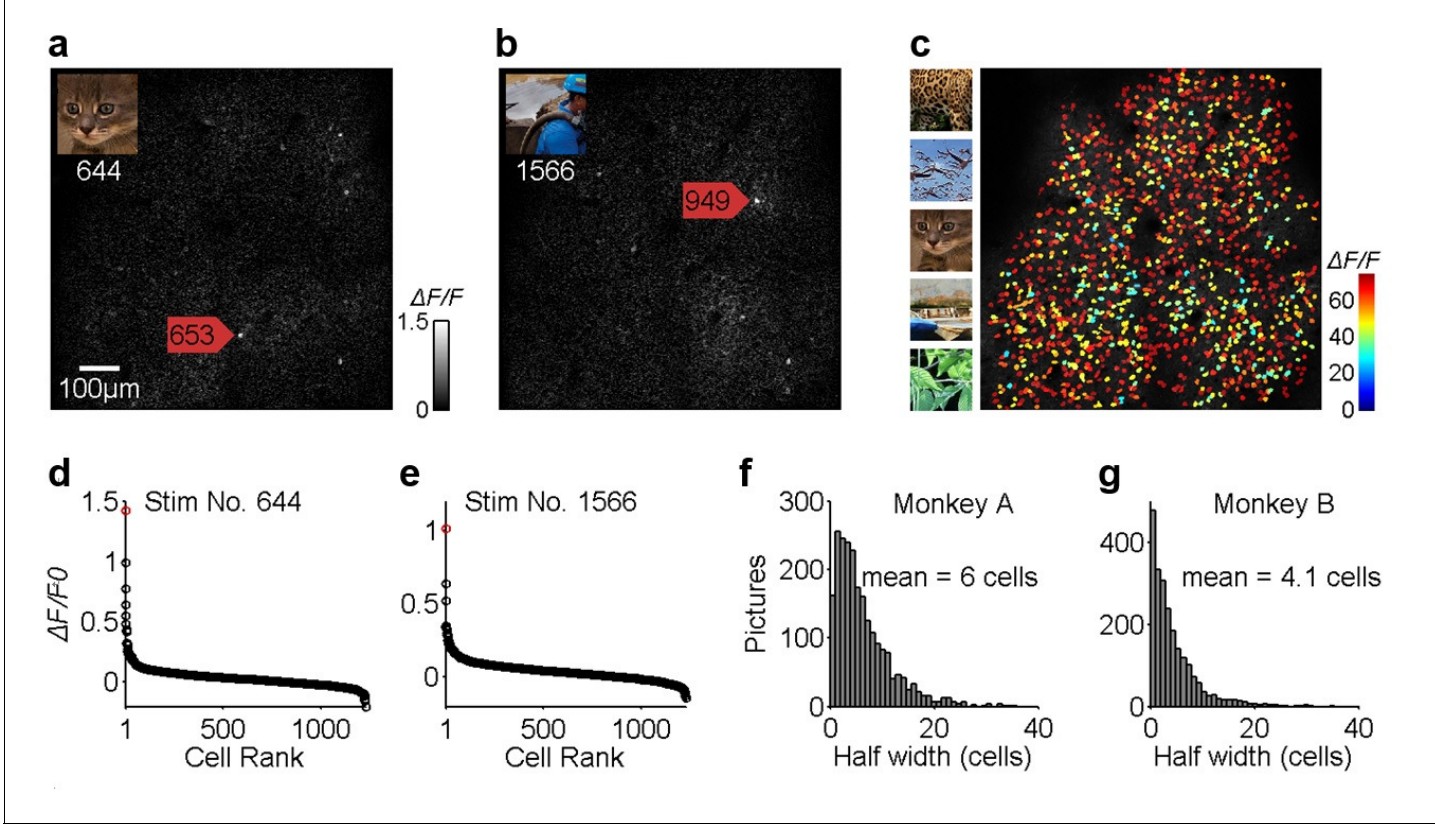

**Figure 1.** Population sparseness of neuronal responses of V1 layer two neurons to natural scenes. (**a, b**) Calcium images of the neuronal population response to two different natural images (shown in the insets). Typically, only a few neurons, among the nearly 1000 neurons measured (1225 neurons for Monkey A or 982 neurons for Monkey B), responded strongly to a single patch of natural scenes. (**c**) The overall neuronal population responses to all 2250 natural images. Each cell was color-coded according to the response intensity to its optimal stimulus. (**d, e**) The distributions of neuronal population responses to the two natural images. Abscissa indicates the 1225 neurons that showed a significant response to natural images, in ranked order according to their responses to each image. Ordinate indicates $\Delta F/F0$. (**f, g**) Frequency histograms showing the distributions of the number of stimuli (out of 2250) (y-axis) with different population sparseness, measured by the number of neurons activated strongly (x-axis). On average, fewer than 0.5% of the cells (6 cells out of 1225 for monkey A, and 4.1 cells out of 982 for monkey B) responded above half of their peak responses for any given image.

DOI: https://doi.org/10.7554/eLife.33370.002

The following figure supplements are available for figure 1:

**Figure supplement 1.** Two-photon calcium imaging in awake macaque monitoring the neuronal activity in V1 layer 2 evoked by natural stimuli.

DOI: https://doi.org/10.7554/eLife.33370.003

**Figure supplement 2.** Two-photon images and neuronal responses in monkey B.

DOI: https://doi.org/10.7554/eLife.33370.004

**Figure supplement 3.** The ROIs overlaid over a two-photon image of a 850 × 850 μm region under a 16X objective, showing that the ROIs extracted on the basis of activities were well matched to the cell bodies.

DOI: https://doi.org/10.7554/eLife.33370.005

Now that we know the highest achievable decoding performance of the entire population, we can build on that idea to assess the contribution of strong sparse responses to the overall decoding accuracy. We zeroed all responses below the top 0.5% response threshold and built a new decoding model to discriminate the 2250 stimuli. Here, we assume that only the strong signals above the threshold would be conveyed to downstream neurons successfully. Remarkably, a decoding accuracy of 28% for monkey A and 21% for monkey B could be achieved with only the top 0.5% of the strongest signals included (*Figure 3a,b* vertical gray lines). This means that transmitting the top 0.5% of the strongest responses for each image was sufficient to realize 50% of the maximum achievable decoding performance for either monkey.

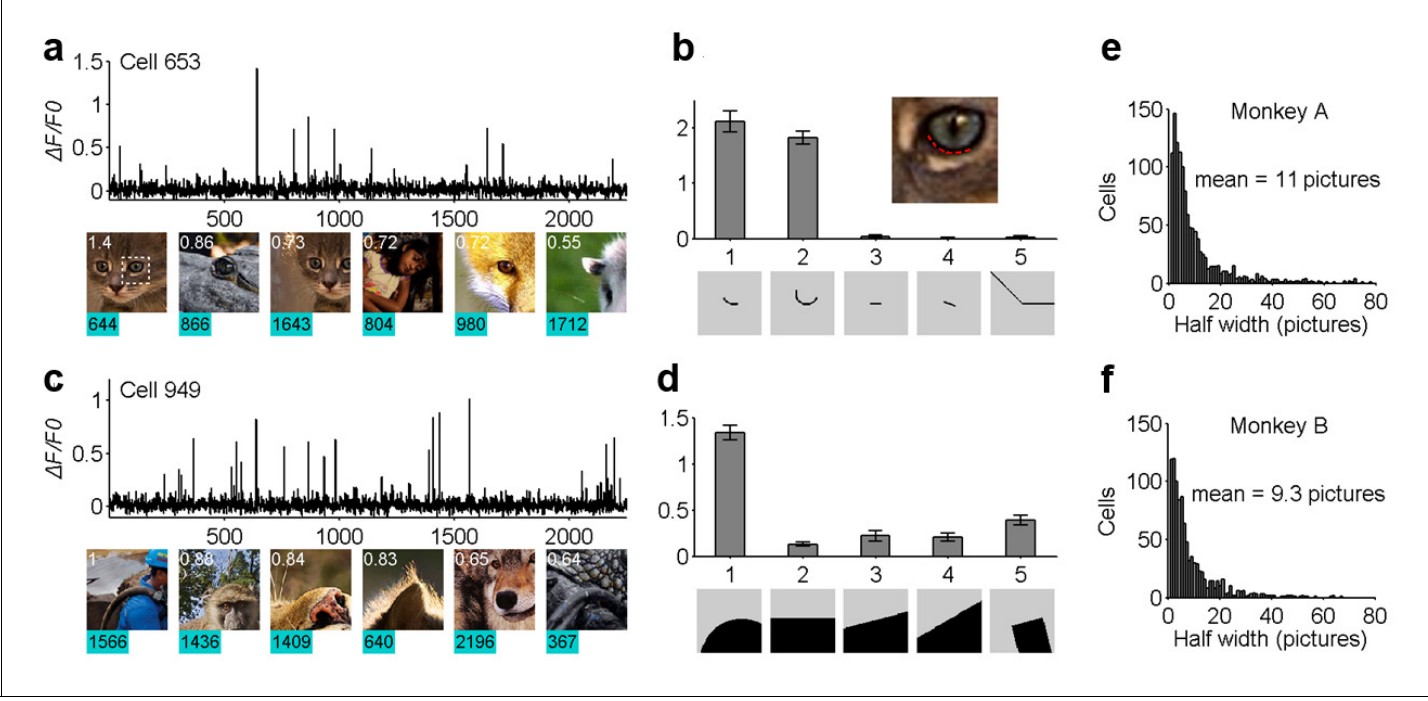

**Figure 2.** Life-time sparseness in the neuronal responses of V1 layer two neurons to natural scenes. (**a, b**) The response of one example cell (cell 653) to the entire set of natural scene stimuli, exhibiting a high level of stimulus specificity. (**c, d**) Another example cell (cell 949) also shows high stimulus specificity. (**e, f**) The distributions of the stimulus specificity of neurons, in terms of the half-height width of the stimulus tuning curves. Each cell would typically respond strongly to fewer than 0.5% of the natural images in our test set.

DOI: https://doi.org/10.7554/eLife.33370.006

The following figure supplement is available for figure 2:

**Figure supplement 1.** Reliability analysis of neuronal responses.

DOI: https://doi.org/10.7554/eLife.33370.007

Conversely, we can assess the necessity of the top 0.5% of responses by repeating the decoding experiment after removing the strongest signals by setting them to 0, and keeping the remaining 99.5% of the signals intact. The blue curves in *Figure 3a and b* show the decoding performance with signals above a range of percentage threshold 'removed'. The intersections of the blue curves with the gray lines indicate that the decoding performance dropped by 50% when the strongest 0.5% of the signals are removed. Thus, we showed that these strong and sparse signals contain the information necessary to realize half of the decoding performance.

*Figure 3a and b* also provides a more complete picture of the decoding performance as a function of percentage threshold. Although the top 0.5% of the signals are both necessary and sufficient for realizing 50% of the performance, 99% of decoding performance is not reached until the top 40% (for monkey A) and top 30% (for monkey B) of the responses are included (saturation of the red curves). However, without the top 5% responses, the decoding performance drops to practically zero (blue curve). Thus, the strongest 5% of the responses are necessary for the achievement of full performance, but insufficient by themselves. In other words, the presence of other weaker signals in the population is required to achieve the full performance. The decoding results revealed that significant information contents are indeed carried by the superb sparse strong responses.

We note that, in the analyses described above, the selection of responses that were kept or removed was based on the absolute response magnitude of the cells, rather than on a normalized response magnitude for each cell as determined by its peak response. We made this decision because we assumed that downstream neurons may know where the signals come from, but it is not clear whether they can know (or remember from historical responses) the peak response magnitude of the neuron that is providing the signals. For better comparison with the population sparseness

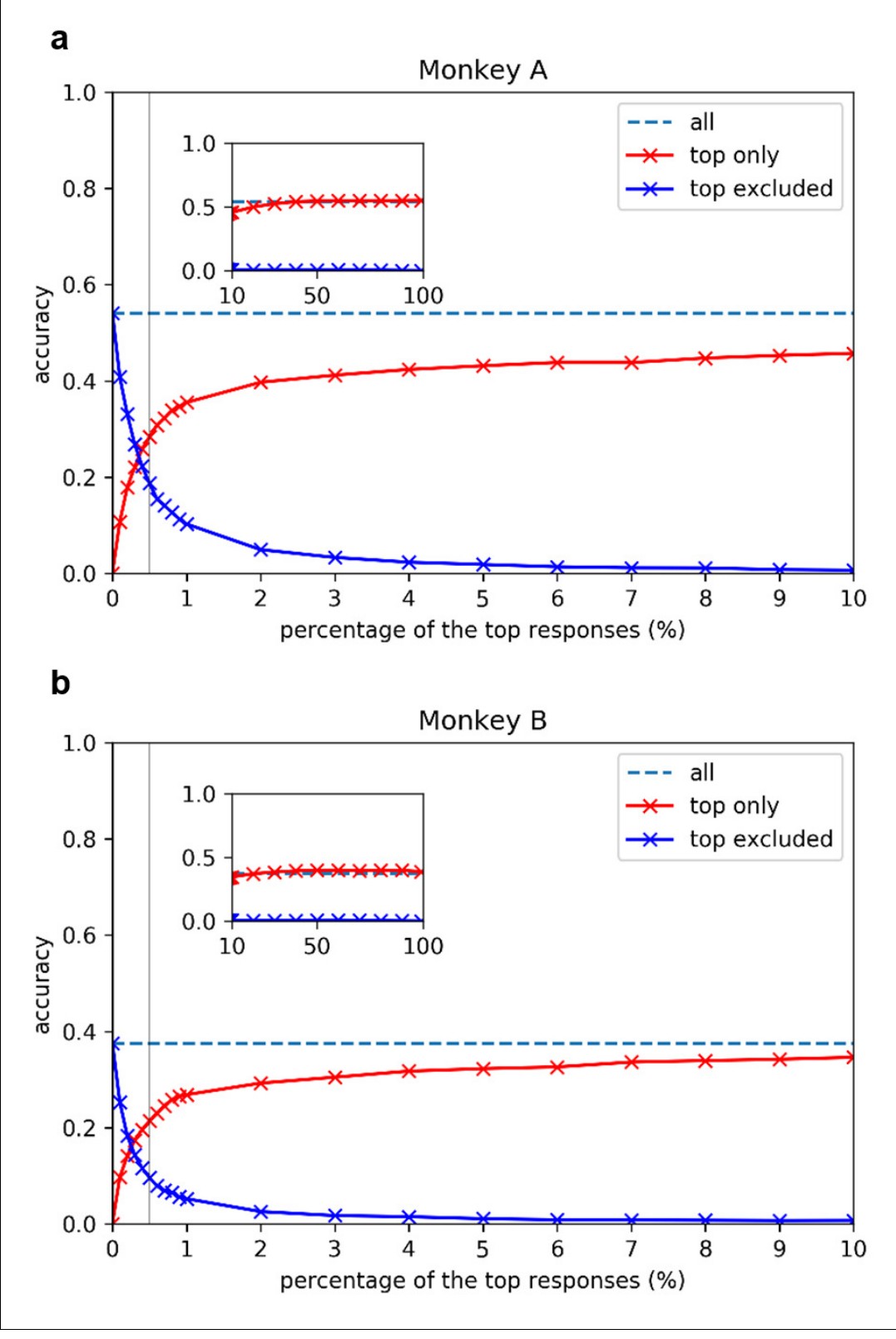

**Figure 3.** Image decoding performance as a function of the percentage of only the strongest responses used for (a) Monkey A and (b) Monkey B. Y axes show the cross-validated decoding accuracy on the 2,250-way image classification task. Dash lines are the referential 'achievable decoding performance' in accuracy obtained using the original entire neural population responses; red lines ('top only' in key) show the decoding accuracies when
*Figure 3 continued on next page*

*Figure 3 continued*

different percentages of the top responses were kept and lower responses were removed (set to zero); blue lines ('top excluded' in legends) show the decoding accuracies when different percentages of top responses were removed (set to zero) and lower responses were kept. X axes show the percentage of top responses included (red curves) and excluded (blue curves). Check 'Materials and methods *Decoding Analysis'* for details. Gray vertical lines highlight the decoding accuracies including or excluding the top 0.5% responses. Since our classification task is a 2250-way one, the chance accuracy is 1/2,250, or about 0.4%.

DOI: https://doi.org/10.7554/eLife.33370.008

The following figure supplement is available for figure 3:

**Figure supplement 1.** Image decoding performance as a function of the percentage of top-responding neurons selected to be included or excluded, using a threshold relative to the peak response of each neuron.

DOI: https://doi.org/10.7554/eLife.33370.009

measure, we repeated the decoding experiment but this time chose the percentage threshold relative to the peak response of each individual neuron. The results were qualitatively similar, and showed that strong sparse responses still carry a disproportionate amount of information (***Figure 3— figure supplement 1***). Quantitatively, the top 1.5% of the responses are now required to achieve 50% of the decoder's highest achievable performance. This suggests that absolute response strengths may potentially convey more discriminable information to downstream neurons. The decrease in performance based on the top 0.5% of responses selected on the basis of a relative threshold (***Figure 3—figure supplement 1***, red curve) is understandable because the relative threshold will include some useless contribution from neurons that have weak peak responses and will exclude the more useful contribution of some neurons that have high peak responses, amplifying the effect of noises particularly when a small number of neurons were selected.

In conclusion, this study provides the first simultaneous recording of a large dense population of neurons in V1 at single-cell resolution in response to a large set of natural stimuli, using 2P imaging in awake macaque. Earlier studies had provided life-time sparseness measurements in rodents (***Hromádka et al., 2008***; ***Haider et al., 2010***), non-human primates (***Rolls and Tovee, 1995***; ***Vinje and Gallant, 2000***; ***Rust and DiCarlo, 2012***) and humans (***Quiroga et al., 2005***), as well as population sparseness measurement in rodents (***Froudarakis et al., 2014***), but our study provides the first direct measurement of sparseness of large-scale neuronal population responses, carried out in awake macaque monkeys and made possible by large-scale 2P imaging techniques. We found that a very small ensemble of neurons from V1's superficial layer are activated at any one time in response to any given natural image. Using decoding analysis, we showed that the small ensembles of neural responses provide a surprisingly large amount of information to downstream neurons, providing for the discrimination of complex image patterns in natural scenes.

Earlier studies inferred population sparseness on the basis of measurements of life-time sparseness. For the first time, we have shown that direct measurements of population sparseness are indeed comparable to life-time sparseness measurements. However, the level of sparseness that we observed (0.5% at half maximal response) was considerably higher than that predicted by the earlier life-time estimates of sparseness, which were based on single unit recording in macaques (***Rust and DiCarlo, 2012***). Studies on rodents have yielded a considerable range of estimates of sparseness that vary across measurement techniques (***Haider et al., 2010***; ***Hromádka et al., 2008***; ***Froudarakis et al., 2014***). A single-unit study that used the cell-attached patch technique might have been the most accurate to date (***Hromádka et al., 2008***), and it showed that neurons were mostly silent in the awake auditory cortex, inferring that less than 2% of the neuronal population showed 'well-driven' responses (>20 Hz response frequency) to natural sounds. Our imaging shows that neurons in the superficial layers of V1 are densely packed and have small cell bodies. Thus, it may not be possible to obtain stable and well isolated single-unit signals over several hours using extracellular recording methods. Our study has reduced the biases inherent to previous extracellular recording studies — in neuronal sampling, in stimulus sampling, and in single-cell isolation — by invoking a response in virtually all of the neurons in a single field of view and in a particular layer by using a large set of natural stimuli.

The high degree of population sparseness that we observed is consistent with two recent conjectures from theoretical neuroscience. First, based on the metabolic costs of spiking, one group posits

that fewer than 1% of the neurons should be substantially active concurrently in any brain area (*Lennie, 2003*). Second, and more importantly, theoretical sparse-coding studies have suggested that because V1 neurons are at least 200 times more abundant than their thalamic input, V1 neurons could be quite specialized in their feature selectivity and thus highly sparse in their population responses (*Olshausen, 2013*; *Rehn and Sommer, 2007*). We have indeed observed this very finding using 2P imaging techniques in the V1 superficial layers (*Tang et al., 2018*). These findings are reminiscent of the highly specific codes exhibited by neurons in the human medial temporal lobes (*Quiroga et al., 2005*), suggesting that many V1 neurons might be akin to highly specific 'grandmother neurons', although they may encode information in the form of an extremely sparse population code. The observed high degree of population sparseness and life-time sparseness are also consistent with our earlier observation that V1 neurons in this layer were tuned to complex patterns with a great degree of specificity (*Tang et al., 2018*, see also *Hegdé and Van Essen, 2007*). These findings reveal the complexity and specificity of feature selectivity and the super-sparse neural representation within a V1 hypercolumn, providing new understanding of the neural codes in the macaque primary visual cortex.

## Materials and methods

**Key resources table**

| Reagent type (species) or resource | Designation | Source or reference | Identifiers | Additional information |
|---|---|---|---|---|
| Strain, strain background (Macaque) | Rhesus monkeys | Beijing Prima Biotech Inc | http://www.primasbio.com/cn/Default | http://www.primasbio.com/cn/Default |
| Recombinant DNA reagent | AAV1.hSyn.GCaMP5G | Penn Vector Core | V5072MI-R | |
| Software, algorithm | Matab 7.12.0 (R2011a) | MathWorks | Matab 7.12.0 (R2011a) | https://www.mathworks.com |
| Software, algorithm | Codes for the decoding analysis and image movement correction | This paper | Codes for the decoding analysis and image movement correction | https://github.com/leelabcnbc/sparse-coding-elife2018 (copy archived at https://github.com/elifesciences-publications/sparse-coding-elife2018) |

All experimental protocols were approved by the Peking University Animal Care and Use Committee, and approved by the Peking University Animal Care and Use Committee (LSC-TangSM-5).

### Subjects

The study used two adult rhesus monkeys (A and B), who were 4 and 5 years of age and weighed 5 and 7 kg, respectively (*Li et al., 2017*). Two sequential surgeries were performed on each animal under general anesthesia and strictly sterile conditions. In the first surgery, a 16 mm hole was drilled in the skull over V1. The dura was opened to expose the cortex, into which 50–100 nl AAV1.hSynap. GCaMP5G.WPRE.SV40 (AV-1-PV2478, titer 2.37e13 (GC/ml), Penn Vector Core) was pressure-injected at a depth of ~500 μm. After AAV injection, the dura was sutured, the skull cap was placed back, and the scalp was sutured. Then the animal was returned to its cage for recovery. Antibiotic (Ceftriaxone sodium, Youcare Pharmaceutical Group Co. Ltd., China) was administered for one week. After 45 days, a second surgery was performed, in which three head-posts were implanted on each animal's skull, two on the forehead and one on the back of the head. A T-shaped steel frame was connected to these head-posts for head stabilization during imaging. The skull and dura were later opened again to explore the cortex. A glass cover-slip (diameter 8 mm and thickness 0.17 mm) was glued to a titanium ring and gently pressed onto the cortical surface. A ring-shape GORE membrane (20 mm in outer diameter) was inserted under the dura. The titanium ring was glued to the dura and skull with dental acrylic to form an imaging chamber. The whole chamber (formed by thick dental acrylic) was covered by a steel shell to prevent breakage of the cover-slip when the animal was returned to the home cage.

## Behavioral task

During imaging, each monkey sat in a standard primate chair with head restraint and performed a fixation task, which involved fixating on a small white spot (0.1°) within a window of 1° for over 2 s to obtain a juice reward. Eye position was monitored with an infrared eye-tracking system (ISCAN, Inc.) at 120 Hz.

## Visual stimuli

Visual stimuli were generated using the ViSaGe system (Cambridge Research Systems) and displayed on a 17' LCD monitor (Acer V173, 80 Hz refresh rate) positioned 45 cm from the animal's eyes. Each stimulus was presented for 1 s after a 1 s blank within a fixation period of 2 s. We estimated the RF sizes and positions of the imaged neurons with small drifting gratings and bars presented at different locations. The RFs were estimated to be 0.2° to 0.8° in size with RF locations between 3° and 5° in eccentricity for both monkeys.

Drifting and oriented gratings were tested to examine the visual responses of imaged neurons (*Li et al., 2017*). Small patches (0.8° in diameter) of gratings with 100% contrast square waves were presented to the center of RFs of imaged cells, with two spatial frequencies (4.0 and 8.0 cyc/deg) at two temporal frequencies (1 and 2 Hz), six orientations, and two directions (30° apart).

A natural stimulus set (NS) of 2,250 4° × 4° stimulus patches extracted from different natural scene photos was used to examine the neuronal responses to natural stimuli. The order of the stimuli was randomized in each session. These stimuli were tested on monkeys A and B, each with at least three repetitions.

## Eye movement control

We analyzed the distribution of eye-positions during stimulus ON periods. The monkeys' fixation during stimulus presentation (from 1 to 2 s in the graph) was stable and accurate. The distribution of eye positions during stimulus presentation, with standard deviations smaller than 0.05°, was significantly smaller than the typical receptive field sizes (ranging from 0.2° to 0.8°) of neurons at 3–5° eccentricities. To examine whether the eye movement made a significant contribution to the distribution of neuronal population responses, we compared the standard deviations (stds) of eye position in different population response classes of neurons: (1) weak responses ($\Delta F/F0$ <0.5), (2) sparse strong responses (one or two cells responded), (3) dense responses (more than ten cells responded). We found no statistically significant differences in the distribution of eye position data in these three classes (*Tang et al., 2018*), indicating that the observed effects were not caused by movement differences. The ROC and decoding analysis (*Figure 1—figure supplement 2*), demonstrating the reliability of the neural responses across trials, confirm that the sparse population responses were repeatedly evoked by stimuli, and not by random eye-movement jitters.

## Two-photon imaging

After a recovery period of 10 days after the second surgery, the animals were trained to maintain eye-fixation. Two-photon imaging was performed using a Prairie Ultima IV (In Vivo) 2P microscope (Bruker Nano, Inc., FMBU, formerly Prairie Technologies) powered by a Ti: Sapphire laser (Mai Tai eHP, Spectra Physics). The wavelength of the laser was set at 1000 nm. With a 16 × objective (0.8 N. A., Nikon), an area of 850 µm × 850 µm was imaged. A standard slow galvonometer scanner was used to obtain static images of cells with high resolution (1024 × 1024). The fast and resonant scan (up to 32 frames per second) was used to obtain images of neuron activity. The images were recorded at 8 frames per second by averaging each 4 frames. Infected cells of up to 700 µm in depth were imaged. We primarily focused on cells that were 160 µm to 180 µm deep, which included a high density of infected cells.

## Imaging data analysis

All data analyses were performed using customized Matlab software (The MathWorks, Natick, MA). The images from each session were first realigned to a template image (the average image of 1000 frames) using a normalized cross-correlation-based translation algorithm, this corrected the X-Y offset of images caused by the relative movements between the objective and the cortex (*Li et al., 2017*).

The cell density was high in superficial V1, and many cell bodies were quite dim at rest. It was difficult to identify these cells directly by eye or by algorithm on the basis of their morphology as captured in static images. We therefore identified ROIs for cell bodies on the basis of their responses. The differential images (which were averaged frames of the ON stimulus period, from which we then subtracted the average of the stimulus OFF period, for each stimulus condition) were first filtered using low-pass and high-pass Gaussian filters (5 pixels and 50 pixels). Notably, these two filters were used solely for ROI identifications. In all further analyses, we used the raw data without any filtering. Connected subsets of pixels (>25 pixels) with average pixel value greater than 3 stds in these differential images were identified as active neuronal ROIs. Note that these 3 stds empirical value was used only for deciding the ROIs of the activated cells, and was not used as a cutoff threshold for measuring neuronal responses (*Figure 1—figure supplement 3*). The ratio of fluorescence change ($\Delta F/F0$) of these ROIs was calculated for each activated cell. $\Delta F = F F0$, where $F0$ is the baseline activity during the blank screen prior to stimulus onset in each trial and $F$ is fluorescence activity in the ROI during stimulus presentation in the trial. A neuropil-correction was performed with an index of 0.7 (*Chen et al., 2013*).

## Sparseness measure

The sparseness measure is used to quantify the peakedness of the response distribution. There are several different definitions of sparseness and corresponding sparseness measures (*Willmore et al., 2011*). One intuitive one for sparse codes, described by *Willmore et al., 2011*, is that "the population response distribution that is elicited by each stimulus is peaked (exhibits population sparseness). A peaked distribution is one that contains many small (approximately zero) magnitude values and only a small number of large values. Thus, a neural code will have high population sparseness if only a small proportion of the neurons are strongly active at any given time." A measure consistent with this intuition is the percentage of neurons that responded strongly, above a certain threshold relative to their peak response. This measure has been used in other studies (*Rust and DiCarlo, 2012*), typically with a half-peak response threshold.

The sparseness measures based on the calculation suggested by *Rolls and Tovee (1995)* and by *Vinje and Gallant (2000)* are popular for quantifying the sparseness of spiking data, but they are very sensitive to measurement noise and uncertain baselines because of nonlinearities and missing responses in the low spiking-rate range (<10 Hz) in calcium imaging (*Li et al., 2017*). The sparseness measure that we used in this study, which is based on the percentage of the cells or stimuli above the half-maximum of each neuron, is much less sensitive to low-level activities (iceberg effect) or baseline fluctuations in the calcium signal.

## Stability and reliability of the neuronal measurements

For each single neuron, we examined whether the sparse strong responses ($\Delta F/F0$ >50% max) observed across the 2250 stimuli were reliable across trials by performing the following ROC analysis (*Quiroga et al., 2005*). We set all the stimuli that produced mean responses greater than 50% of the observed maximum mean peak of the cell to be in the ON class, and all other stimuli to be in the OFF class. We computed the ROC for classifying the ON class against the OFF class based on the response of each single trial. If the responses above the half-maximum are stable across all trials, then the AUC (the area under the ROC curve) will be close to 1.0, because the ON and OFF classes are readily discriminable. The null hypothesis is that sparse strong responses will be spurious single-trial epileptic responses, and not repeatable across trials. To test this hypothesis, we shuffled all the responses against the stimulus labels, and recomputed the mean responses for all the stimuli across the trials. We performed 1000 shuffles. We found that most of the shuffled cases have much lower average peak responses because of the mismatch of the rigorous sparse responses across trials, suggesting that the sparse responses in the original data are reliable. To make an even stricter and more fair comparison with the original data on ROC terms, for each shuffle, we recomputed the maximum responses, and used the half of this mean maximum as the threshold to sort the stimuli into ON and OFF classes and repeated the ROC analysis to obtain the AUC for this shuffle. The probability of the null hypothesis is the percentage of the time that the AUCs of the 1000 shuffles reach the AUC of the original data. With this ROC analysis, we found >96% neurons that adhered to the null hypothesis (p<0.01) (*Figure 1—figure supplement 2*).

## Decoding analysis

We used a nearest centroid classifier to discriminate the 2250 images on the basis of the population responses in each trial. As each image was tested three times, the nearest centroid classifier was trained on the basis of two trials for all images and tested on the hold-out trials. We repeated the procedure for each trial, performing 3-fold cross-validations.

For each monkey, we constructed neural response matrices (with dimension $2250 \times 1225$ for monkey A, and $2250 \times 982$ for monkey B) for three trials $X^{(1)}$, $X^{(2)}$, and $X^{(3)}$ that store the neural responses to all images in each trial as rows in its matrix. We trained and tested nearest-centroid classifiers via a three-fold cross-validation procedure across trials in a 2250-way image decoding task. Specifically, for trial t, during training, we computed the centroids of the other two trials $C^{(t)}$ (if t = 1, $C^{(1)} = (X^{(2)} + X^{(3)})/2$; if t = 2, $C^{(2)} = (X^{(1)} + X^{(3)})/2$, etc.) and stored $C^{(t)}$ in the classifier; during testing, given some row k of $X^{(t)}$, which is the population neural response vector to image k in trial t, the (trained) classifier computed the Euclidean distances between row k of $X^{(t)}$ and every row of $C^{(t)}$. The model outputted the index (1,2,...,2249,2250) of the row in $C^{(t)}$ that gives the smallest distance. The correct output is k and all other outputs are incorrect. The average decoding accuracy for this trial is defined as the percentage of correct outputs over all rows of $X^{(t)}$. We repeated the above procedure for each trial and reported the average of three (average) decoding accuracies.

In our experiments, we first set the $X^{(t)}$s defined above to be the original recorded neural responses and computed the decoding accuracies for both monkeys. We refer to the accuracies obtained from the original neural data as 'achievable decoding accuracies'. Later, to evaluate the amount of information in the strong sparse portions of the neural data, we set $X^{(t)}$s to be thresholded versions of the original data. We tried two classes of thresholding methods: 'top only' (red in *Figure 3*) and 'top excluded' (blue in *Figure 3*). In 'top only', we only kept the largest responses (p %) across images and trials in the thresholded version and set the smaller responses (100-p%) of the to be zero. In 'top excluded', which is complementary to 'top only', we set the largest responses (p %) to be zero and kept the smaller responses (100-p%). For both 'top only' and 'top excluded', we evaluated decoding accuracies at the following percentages (crosses in *Figure 3*): 0, 0.1, 0.2, 0.3, 0.4, 0.5, 0.6, 0.7, 0.8, 0.9, 1, 2, 3, 4, 5, 6, 7, 8, 9, 10, 20, 30, 40, 50, 60, 70, 80, 90, and 99.

Given that the population sparseness was computed on the basis of the half-maximum of each individual neuron's response, we also repeated the decoding experiment using a percentage threshold that is relative to each neuron's peak response, rather than the absolute response threshold, to select the 'top responding' neurons to be included or excluded and reset the responses to be excluded in the data matrix to 0 accordingly for training and testing the decoder as before.

## Software available

The code used for the decoding analysis and image movement correction can be found in https://github.com/leelabcnbc/sparse-coding-elife2018 (*Zhang et al., 2018*; copy archived at https://github.com/elifesciences-publications/sparse-coding-elife2018).

## Acknowledgements

We are grateful to many colleagues for their insightful discussion and generous help on this paper, and in particular to Stephen L Macknik, Susana Martinez-Conde and Shefali Umrania for editing the manuscript. We thank Wenbiao Gan for the early provision of AAV-GCaMP5; and Peking University Laboratory Animal Center for excellent animal care. We acknowledge the Janelia Farm program for providing the GCaMP5-G construct, specifically Loren L Looger, Jasper Akerboom, Douglas S Kim, and the Genetically Encoded Calcium Indicator (GECI) project at Janelia Farm Research Campus Howard Hughes Medical Institute.

# Additional information

## Funding

| Funder | Grant reference number | Author |
|---|---|---|
| National Natural Science Foundation of China | 31730109 | Shiming Tang |
| National Natural Science Foundation of China | China Outstanding Young Researcher Award 30525016 | Shiming Tang |
| National Basic Research Program of China | 2017YFA0105201 | Shiming Tang |
| Peking University | Project 985 grant | Shiming Tang |
| Beijing Municipal Commission of Science and Technology | Z151100000915070 | Shiming Tang |
| NIH Office of the Director | 1R01EY022247 | Tai Sing Lee |
| National Science Foundation | CISE 1320651 | Tai Sing Lee |
| Intelligence Advanced Research Projects Activity | D16PC00007 | Tai Sing Lee |

The funders had no role in study design, data collection and interpretation, or the decision to submit the work for publication.

## Author contributions

Shiming Tang, Conceptualization, Resources, Data curation, Software, Funding acquisition, Investigation, Methodology, Writing—original draft, Project administration, Writing—review and editing; Yimeng Zhang, Data curation, Writing—original draft, Writing—review and editing; Zhihao Li, Data curation, Investigation; Ming Li, Fang Liu, Hongfei Jiang, Methodology; Tai Sing Lee, Conceptualization, Data curation, Funding acquisition, Investigation, Writing—original draft, Writing—review and editing

## Author ORCIDs

Shiming Tang (iD) http://orcid.org/0000-0003-0294-3259
Yimeng Zhang (iD) http://orcid.org/0000-0003-2248-8951

## Ethics

Animal experimentation: Animal experimentation: All procedures involving animals were in accordance with the Guide of Institutional Animal Care and Use Committee (IACUC) of Peking University Animals, and approved by the Peking University Animal Care and Use Committee (LSC-TangSM-5). All surgrey was performed under general anesthesia and strictly sterile conditions, and every effort was made to minimize suffering.

## Decision letter and Author response

Decision letter https://doi.org/10.7554/eLife.33370.014
Author response https://doi.org/10.7554/eLife.33370.015

# Additional files

## Supplementary files

• Transparent reporting form
DOI: https://doi.org/10.7554/eLife.33370.010

## Data availability

All data generated or analysed during this study are included in the manuscript and supporting files. Source data files have been provided for Figures 1, 2 and 3.

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
