## [Decision Letter]

Thank you for submitting your article "Large-scale two-photon imaging revealed super-sparse population codes in V1 superficial layer of awake monkeys" for consideration by *eLife*. Your article has been reviewed by three peer reviewers, and the evaluation has been overseen by a Reviewing Editor and David Van Essen as the Senior Editor. The following individuals involved in review of your submission have agreed to reveal their identity: Anna Wang (Reviewer #2); Stanley Klein (Reviewer #3).

The reviewers have discussed the reviews with one another and the Reviewing Editor has drafted this decision to help you prepare a revised submission.

Summary:

In this study, Tang and colleagues use two-photon calcium imaging in awake monkeys to investigate the sparseness of responses in V1. All reviewers were enthusiastic about the manuscript, noting that it was technically novel and conceptually important for delineating how V1 encodes the visual world. For instance, 5 important advances were pointed out: (1) No previous study has provided a direct data-based quantitative measure of sparseness in terms of number of cells in the population (<0.5%). This finding demonstrates that very few cells, for a large number of stimulus types (from simple to complex), are needed to encode each stimulus. This is a conclusion that, due to limitations of sampling, could not be arrived at by existing single unit neurophysiology approaches. (2) The study presents a large number of different stimuli, both simple and complex, to a large dense population of single cells, all observed simultaneously in monkey cortex. This is an innovative way to track how all cells within a single locus of cortex respond under a large number of different conditions. (3) This study invites reconsideration of V1 as an area that encodes simple features of visual stimuli. (4) This study opens up new ideas about what a hypercolumn is. (5) Technically, this is a high quality, tour-de-force study of monkey visual cortex that few in the world can achieve.

The reviewers also pointed out several areas where the manuscript requires improvement, most notably (1) discussing the possibility that GCaMP is missing lower firing rate neurons; (2) improving the clarity and justification of the analysis methods, as well as possible alternate interpretations of current findings; (3) improving the discussion of why these findings are novel, and what are the implications for the role of V1 in visual processing, as well as for the meaning of a hypercolumn; and (5) making the figures more clear. Specific comments and suggestions follow.

Essential revisions:

1) Concerns about GCaMP5.

1a) The authors point to the linearity of GCaMP5 as an advantage of the current study. While this is correct for large enough firing rates, it fails to mention the significant iceberg effect of GCaMP5, which is clearly demonstrated in the authors previous work (only spike rates larger than 10Hz evoke detectable fluorescence changes). In the current paper, the iceberg effect is only acknowledged in passing in the Materials and methods section. This iceberg effect is a serious issue for the current study. This paper reports average firing rates between 15 and 30 Hz, which would suggest that a detection threshold of 10Hz for the calcium imaging indeed will result in an overestimation of sparseness by failing to detect weaker responses. Most of the chronic calcium indicators currently available (including GCaMP6) have this problem, so there is currently no suitable alternative. However, the shortcomings of the chosen indicator, and the resulting potential for overestimating sparseness, should be addressed much more clearly than is currently the case. In its current form the paper is misleading, because it does not acknowledge the confounds of the GCaMP nonlinearities on the sparseness measurements.

1b) One fix for this problem could be to include actual measurements of spike rates for a population of V1 neurons. While those data would suffer from sampling biases not present in the two-photon data, they would still be the strongest possible complement to this data set. Without spike data, it is basically impossible to assess how much of a problem the iceberg effect could be.

1c) Perhaps a sensitivity analysis could be performed to test different assumptions about the response distributions below the Ca signal threshold. For instance, what happens if each "non-responsive" neuron is assigned a random response between 0 and 10 spikes/s? How would the sparseness results change in that case? That type of statistical analysis could be useful to estimate or provide bounds to the error in the sparseness measurement caused by the iceberg effect.

1d) The statement that the linearity of GCaMP5 makes the sensor more suitable than GCaMP6 (which potentially saturates at higher rates) is incorrect. Determining sparseness across a population requires a determination of how many neurons are 'on' during a particular stimulus presentation, not accurate measurements of tuning functions. The same holds for determining the life-time sparseness of a neuron, which also only requires a determination of how many stimuli drive a neuron over a baseline level, not the precise tuning function.

2) Concerns about statistical/analysis methods

2a) Sparseness measures. The authors use the half-height bandwidth as a measure for sparseness. This choice is rather arbitrary and should be further justified. It would seem more plausible to count all stimuli that evoke responses that are significantly larger than the baseline. At a minimum, the authors should explore how their assessment of sparseness changes if the criterion threshold is changed (how many stimuli evoke responses that are 10%, 20% etc. of the maximum response). In general, any such measure is problematic because of the iceberg effect of GCaMP mentioned above. This also needs to be discussed more explicitly.

2b) Comparison to traditional sparseness measures. The authors assert that sparseness measures used previously (as the one by Rolls and Tovee) are not applicable here because they are sensitive to changes in baseline level. However, these previous studies used baseline subtracted firing rates to calculate sparseness. The sensitivity of the traditional measures to changes in baseline levels therefore requires further explanation.

2c) Decoding. The 2 analysis parts of the paper are somewhat disconnected. One emphasizes single cell selectivity, whereas the other emphasizes population sparseness. It might be useful to set up the idea that single cell selectivity does or does not predict population sparseness. It seems two concepts are correlated, but I could imagine that this need not be so.

The first part assesses sparseness by thresholding responses on a per neuron basis. The second part assesses decoding based on groups of neurons by thresholding the population. What happens to the decoding if the response matrix is computed with the thresholding of part 1 applied (i.e., setting all responses below the half maximum for a neuron to 0)?

Furthermore, the discussion of the decoding results should be improved. Currently, it seems to imply a rather arbitrary threshold of around 20% that is considered 'good decoding' (e.g., in the comparison of the decoding results from the top 0.5% – which are around 20-30%, and the decoding results from the bottom 0.5% – which are around 15%). Both are far from the chance level, so these statements need to be further justified.

Finally, the authors conclude that the comparison of decoding performance for top and bottom responses demonstrates that strong responses are both necessary and sufficient to transfer relevant information. This is incorrect. The sufficiency is indeed demonstrated (accepting the assertion that decoding performance above some threshold constitutes successful decoding). However, to demonstrate necessity, they would have to demonstrate that successful decoding always implies the occurrence of strong responses. This is not the same as demonstrating that weak responses do not allow 'successful' decoding.

2d) It would be interesting to compare the sparseness of responses evoked by natural images to that evoked by gratings (which in other studies have been shown to drive a large percentage of superficial V1 neurons). This would also allow a better assessment of how many neurons could potentially respond, further alleviating concerns about cell health or other properties of the imaging region (although this concern is largely addressed by the fact that most neurons respond to at least some of the images in the natural image set).

2e) A famous paper by Quiroga, Reddy, Kreiman, Koch and Fried (2005) illustrated extremely high sparsity in cells that responded to images of Jennifer Aniston and Halle Berry. I'm actually surprised that there was no mention in the present paper of that finding by Quiroga. Although a direct comparison may not be appropriate given the differences in areas, it may still be informative to ask whether the V1 cells have greater sparseness than the Jennifer Aniston cell.

Another reason for connecting to the Quiroga paper is that they also do ROC analyses, but their ROC curves look very different than those of the present paper (see point 5e below). The comparison may provide further evidence that the sparseness calculations, including the ROC calculations, were done properly.

3) Clarification of experimental methods employed.

3a) Cell count. The overall number of cells per imaging region is crucial for estimating sparseness. The ROI-definition procedure adopted by the authors appears reasonable and well justified. However, a few additional details would be useful:

- Additional images of identified cells so that the accuracy of the chosen approach can be assessed.

- How does the imaging region from monkey 2 look like?

- Which manual steps are involved in the procedure (presumably, somebody checks the identified ROIs)?

- Were all data collected on the same day? If not, how are data and cell counts combined across days? How many cells were stable across those days?

- How many of these cells are filled in versus have a ring-like label? This will help to assess how many of the included neurons are presumably healthy and should exhibit normal responses.

3b) Visual stimuli. Only the size of the stimuli are given. What are the other characteristics of the natural image set? Are they in color? Are they isoluminant with the background? What spatial frequencies and colors do they span? How different is their content? Are they part of one of the standard sets of natural images used in other studies?

The claim that single neurons respond to similar features of stimuli is not well supported and premature [“neuron 653 of monkey A was most excited when its receptive field (0.8o in diameter) covered the lower rim of the cat's 1 eye'. 'neuron 949 of Monkey A was found to be selective to an opposite curvature embedded in its preferred natural stimulus set”].

4) Points for further discussion/elaboration.

4a) The Discussion is focused on issues of sparseness. However, there are two issues that are also worth mentioning. First, this study thus provides a fresh view of what V1 is doing, one that shifts the emphasis from simple orientation selectivity to complex natural stimuli, and which gives novel perspective on how the brain encodes natural stimuli. And second, the results show that, within the span of a single hypercolumn, all stimuli presented could be largely decoded. This supports Hubel and Wiesel's original concept that a hypercolumn contains all the machinery to encode everything at a single point in space, except that the manner of encoding may be distinct from (or more complex than) the original concept of selection from amongst an array of systematically organized ocular dominance, orientation, and color columns.

4b) A very important item that should be made very clear is the last sentence of the Abstract where the authors correctly claim that this is the first paper that shows sparseness of neuronal population responses in V1 in primates. They need to point out that papers like Frouderakis et al. (2014) were in mice and papers like Quiroga, et al. (2005) were in humans but not in V1. The statement at the end of the tenth paragraph of the Results and Discussion needs to make that clear. It is such an important point that it needs to be pointed out in the Introduction and the Conclusion. Other researchers would then become aware of the power of two-photon imaging.

5) Clarification of data/analyses in the figures.

5a) I find Figure 1F and G vs. L and M description very confusing. I think this is the interpretation: For population sparseness, one expects the distribution to show that for most images only few cells respond. For single cell selectivity, one expects each cell to respond to only a few images. Somehow the description of these graphs seems garbled. [E.g. then for each picture only 0.5% of cells responded means: in Monkey A, for 250 pictures 1 cell responded; for ~200 pictures 6 cells responded; for 5 pictures 20 cells responded. Alternative interpretation of graph: 1 cell responded to 250 pictures; 6 cells responded to ~200 pictures; 20 cells responded to 5 pictures? I would not use 'half-height bandwidths'. Use 'number of cells'. Why is stimulus specificity called 'life-time sparseness'? F-G, L-M should be described better to distinguish what each is saying (single cell selectivity vs. population sparseness/redundancy). Maybe partly it is the terminology that is used.]

5b) Figure 1: Why are there points below zero in D and E?

5c) Figure 1: 'Cells 653 and 949 are colored red respectively.' Don't see this.

5d) Figure 2: 0.5% contributes 50% of information and 5% contributes to 80% of information, what are remaining cells doing?

5e) Figure 1—figure supplement 2: Item C shows ROC curves for 99 shuffled trials. The part that wasn't at all clear to me was why did you compare the results to a shuffled version of the results. And why did the shuffled data have such a high hit rate at zero false alarms. I would have thought that the shuffling would greatly reduce the hit rate. That is quite different from Quiroga's (2005) paper, which shows more normal curves with the false positive rate close to zero. If the authors are unable to use the Quiroga method then they should explain why, and why they end up with the very unusual shape for the ROC curves.

5f) Perhaps it would be helpful to the reader to have Figure 1 broken up into multiple figures. By doing that the figures would be larger with details more visible. One improvement that would be helpful for Figures1D, E would be to have the x-axis be a log axis so that the cell rank of the first 20 or so neurons would be more visible. I believe this is the plot that best demonstrates sparseness so it needs to be very clear.

I would suggest also spending substantially more effort clarifying F and F0 on which panels D and E are based. Another question is what is the role of noise for the one second intervals where the stimulus is shown? There is expected to be more noise in that interval than in the interval without the stimulus. How is that noise estimated?

5g) In connection with Figure 1—figure supplement 2 it would be useful to show A and B as histograms in addition to the way they are presently shown.

5h) I suspect Figure 2 is an important figure. However it wasn't at all clear to me how it was calculated. I did figure out that the small inset was a continuation of the large plot that stopped at 10%. I wonder of the inset could be removed if log axes were used for the x-axis and the data would go from say 0.1% to 100%. But more important is to clarify how the red and blue data were calculated.

---

## [Author Response]

Essential revisions:1) Concerns about GCaMP5.1a) The authors point to the linearity of GCaMP5 as an advantage of the current study. While this is correct for large enough firing rates, it fails to mention the significant iceberg effect of GCaMP5, which is clearly demonstrated in the authors previous work (only spike rates larger than 10Hz evoke detectable fluorescence changes). In the current paper, the iceberg effect is only acknowledged in passing in the Materials and methods section. This iceberg effect is a serious issue for the current study. This paper reports average firing rates between 15 and 30 Hz, which would suggest that a detection threshold of 10Hz for the calcium imaging indeed will result in an overestimation of sparseness by failing to detect weaker responses. Most of the chronic calcium indicators currently available (including GCaMP6) have this problem, so there is currently no suitable alternative. However, the shortcomings of the chosen indicator, and the resulting potential for overestimating sparseness, should be addressed much more clearly than is currently the case. In its current form the paper is misleading, because it does not acknowledge the confounds of the GCaMP nonlinearities on the sparseness measurements.

The concerns about iceberg effect may have arisen due to the fact that we may not have sufficiently clarified the definition of sparseness we used. There are in fact several different definitions of sparseness rather than one. We used a definition of sparseness that is based on quantifying the peakedness of neuronal response distribution, which is robust and less sensitive to low level activities. The iceberg effect is not a serious problem in the sparseness measure (based on this definition we used) in this study.

1) Willmore, Mazer and Gallant, 2011 discussed four definitions of sparseness commonly used in the field. One we found intuitive is as follows: “a sparse code (is defined) as one in which the population response distribution elicited by each stimulus is peaked (we will refer to this as “population sparseness”). A peaked distribution is one that contains many small (or 0) values and a small number of *large* values. Thus a neural code will have high population sparseness if only a small proportion of neurons are *strongly active* at any given time”.

2) The half-height width of the ranked response distributions is a simple measure of the peakedness. A small half-height width of the ranked response distributions indicates “only a small proportion of neurons are strongly active at any given time”, Thus a half-height width of the ranked response distributions has provided a standard and comparable measure of sparseness (Rust and DiCarlo, 2012).

3) There are several different definitions of sparseness (and corresponding sparseness measures) rather than one. Actually, there is no simple perfect measure of sparseness. The popular sparseness measure suggested by Rolls and Tovee could not tell precisely the peakedness of a neuronal response distribution because multiple response distribution profiles can give the same sparseness index. For example, consider the following two neurons, one neuron has tens of strong peak responses and hundreds of very low responses (i.e. fat head but thin tail in its ranked response distribution), while the other neuron has only a few strong responses and hundreds of medium low responses (thin head but fat tail in the distribution). These two neurons have quite different response distributions, but could have same Rolls and Tovee’s sparseness measure. That means, when one study reports a low sparseness based on Rolls and Tovee’s measure, we will have no way of knowing whether this low sparseness comes from the dense peak responses (the fat heads) or medium weak responses (the fat tails). Similar issue is also true for the population sparseness measure.

4) The sparseness definition suggested by the reviewers – “a determination of how many neurons are 'on' (over baseline)” – has problems too. It would have treated the peaked strong responses and distributed weak responses equally, and thus could not characterize the peakedness of the response distribution. Probably because of these defects, a simply “on” definition of sparseness, as suggested, is not popularly used in this field.

5) Balancing all considerations, in this study, we used a half-height width of the ranked response distributions (which is the percentage of neurons exceed their half peak responses – for population sparseness, and the percentage of stimuli that makes a neuron fire more than half of its peak response – for life-time sparsity) to quantify the peakedness of response distribution. This measure is robust and less sensitive to variations in low-level activities (including the noises and nonlinearity in low firing range in calcium imaging). To make this issue more explicit, we add a statement in the text that this sparsity measure is not just indicating the percentage of neurons exhibit any statistical significant responses (probably a large fraction of neurons will show some weak responses, and given enough trials, the responses can be deemed significant), but the percentage of neurons exhibit strong response (above half their max firing rates). Half-height is used to decide whether a response is “strong” or not for computing sparseness. The half-height bandwidth of ranked response distribution has been used as a sparseness measure in other studies (Rust and DiCarlos), and is not an arbitrary measure we developed.

6) As explained above, the sparseness measure based on the half-height width of the ranked response distributions (the peakedness measure) concerns mostly the strong responses (mountains) but not on the distributed weak responses (valleys) or mean firing rates. Notably, a highly selective neuron with very strong responses to some optimal stimuli (which may exceed 150 Hz) will likely have very low average firing rate (even under 10 Hz). We have found, as reported in our Neuron paper, the firing rates below 10 Hz (~0.1 in dF/F0) could not linearly measured by GCaMP5. Missing these low level responses would surely under-estimate how many neurons would turn on, as the reviewer rightly pointed out. But, given the mountains of strong responses is far above that sea-level of 0.1, its half-height is also way above the sea-level, and the iceberg effect is quite minimal for the half-height bandwidths sparseness measure used in this study (see also discussion below 1c).

1b) One fix for this problem could be to include actual measurements of spike rates for a population of V1 neurons. While those data would suffer from sampling biases not present in the two-photon data, they would still be the strongest possible complement to this data set. Without spike data, it is basically impossible to assess how much of a problem the iceberg effect could be.

It is true that obtaining a distribution of spiking rates of a population of V1 neurons to the 2,250 stimuli would be a good comparison. However, this is not a simple task. First, as the reviewers pointed out, it is really hard to densely sample neurons with microelectrodes without bias. Second, the neurons in that layer are densely packed which would make isolation and cell-sorting difficult. Third, chronic multi-electrode recording of a particular superficial layer might not be stable enough to obtain the equivalent data. Given these factors, whatever comparative studies would likely not be conclusive. Furthermore, given the findings of our Neuron paper, and with a better clarification of our sparseness measure and its rationales, we feel such comparison is not completely necessary, particularly in view of the fact that our sparseness measure has been chosen precisely to mitigate the iceberg or sea-level effect that the reviewers were worried about.

1c) Perhaps a sensitivity analysis could be performed to test different assumptions about the response distributions below the Ca signal threshold. For instance, what happens if each "non-responsive" neuron is assigned a random response between 0 and 10 spikes/s? How would the sparseness results change in that case? That type of statistical analysis could be useful to estimate or provide bounds to the error in the sparseness measurement caused by the iceberg effect.

We have performed this test according to reviewers’ suggestion. We assigned each non-responsive neuron (with dF/F0 < 0, also tested with dF/F0 < 0.1) with random responses from 0 to 0.1 dF/F0 (roughly corresponding to 0 to 10 spikes/s). Exactly as discussed above, we found that the iceberg effect on sparseness measure is very small. In most cases, the half-height widths of ranked responses barely changed. In monkey A, the measured sparseness slightly increased from 0.49% to 0.50%. In monkey B, the measured sparseness did not change at all (0.41%).

1d) The statement that the linearity of GCaMP5 makes the sensor more suitable than GCaMP6 (which potentially saturates at higher rates) is incorrect. Determining sparseness across a population requires a determination of how many neurons are 'on' during a particular stimulus presentation, not accurate measurements of tuning functions. The same holds for determining the life-time sparseness of a neuron, which also only requires a determination of how many stimuli drive a neuron over a baseline level, not the precise tuning function.

We disagreed with the reviewers on this point. The reviewers might have missed a crucial aspect in the definition of sparsity, which does depend on the shape of the tuning curves. As mentioned above, we used an original definition of sparseness: “a neural code will have high population sparseness if only a small proportion of neurons are strongly active at any given time”. This definition is also applicable to life-time sparseness as well. We have stated that our sparseness measure is the percentage of *strong* response, not simply just any significant response. We will make sure this point clearer in the main text.

The advantage of GCaMP5 is that it catches all the peaks in responses even though it misses some low-level activities (below 10 Hz). GCaMP6s is more sensitive and tends to saturate above 60-80 Hz, and flattening many strong peaks into plateau, reducing sparseness measure, although it can capture more weak responses in the low-firing rate regime. Thus GCaMP5, not GCaMP6, is more appropriate for this study. However, if one’s definition of sparseness is simply the percentage of neurons would be turned on (regardless strength), then GCaMP6 would be more appropriate. But as discussed above, such a definition, while seems intuitive to general audience, does not characterize the peakedness of the response distribution, which is a critical aspect of the preferred definition of sparseness in the field (see Willmore et al., 2011).

2) Concerns about statistical/analysis methods2a) Sparseness measures. The authors use the half-height bandwidth as a measure for sparseness. This choice is rather arbitrary and should be further justified. It would seem more plausible to count all stimuli that evoke responses that are significantly larger than the baseline. At a minimum, the authors should explore how their assessment of sparseness changes if the criterion threshold is changed (how many stimuli evoke responses that are 10%, 20% etc. of the maximum response). In general, any such measure is problematic because of the iceberg effect of GCaMP mentioned above. This also needs to be discussed more explicitly.

As discussed above (1a), we did not “count all stimuli that evoke responses that are significantly larger than the baseline” or “a determination of how many neurons are 'on' (over baseline)”, as a definition of sparseness because measures based on such definition treat the peaked strong responses and distributed weak responses equally and could not characterize the peakedness of the response distribution, which is a critical aspect of the commonly used sparseness definitions sparseness in the field (see Willmore et al., 2011).

Denser sampling of the bandwidths of the ranked response curve at various response height thresholds (such as 75%, 50%, 25% and so on) might provide a more comprehensive profile of the responses distributions, as shown in Author response image 1. However, a half-height width of the ranked response distributions is commonly used for counting strong responses and quantifying the peakedness. A very low height threshold will not work in quantifying the peakedness. The sparseness measure based on the peakedness analysis is not sensitive to low level iceberg effect (see also the response in (1c)).

**Author response image 1. respfig1:** The proportions of stimuli those evoked strong responses above varying thresholds.

2b) Comparison to traditional sparseness measures. The authors assert that sparseness measures used previously (as the one by Rolls and Tovee) are not applicable here because they are sensitive to changes in baseline level. However, these previous studies used baseline subtracted firing rates to calculate sparseness. The sensitivity of the traditional measures to changes in baseline levels therefore requires further explanation.

Precisely, Rolls and Tovee’s measure requires the subtraction of a baseline. However, determination of a baseline precisely is difficult for Ca imaging signals. As we tested thousands of stimuli, a slight variation in our baseline estimate will yield a large variation in the estimate sparseness value, because of most of the responses are very weak, and there is the danger of iceberg effect, hence, we chose a measure that does not require a precise estimation of baseline and is more robust against the known insensitivity against signals below 10 Hz, as discussed above. We will make further clarification in the Materials and methods section.

2c) Decoding. The 2 analysis parts of the paper are somewhat disconnected. One emphasizes single cell selectivity, whereas the other emphasizes population sparseness. It might be useful to set up the idea that single cell selectivity does or does not predict population sparseness. It seems two concepts are correlated, but I could imagine that this need not be so.

Actually, the first analysis part includes both population sparseness (Figure 1) and stimulus selectivity (life-time sparseness) (Figure 2). A paper from Willmore, Mazer and Gallant, 2011 discussed the correlation between lifetime sparseness and population sparseness carefully. Simply, when the responses are distributed (like those in this study), these two sparseness measures will be quite close. The second part is about decoding. The main motivation for the decoding analysis is to examine the amount of information that is contained in the strong sparse responses.

The first part assesses sparseness by thresholding responses on a per neuron basis. The second part assesses decoding based on groups of neurons by thresholding the population. What happens to the decoding if the response matrix is computed with the thresholding of part 1 applied (i.e., setting all responses below the half maximum for a neuron to 0)?

This is a correct observation. We agreed that there is certain dissonance in using relative half-height threshold in charactering population sparseness in Part 1, and then using absolute threshold for doing decoding in Part 2. We have actually done both, but we feel that using absolute threshold makes it more clear that the strong response in the absolute sense is important, as there is no guarantee that a cell can necessarily attain its absolute peak responses among these 2,250 natural image set. While decoding with absolute response threshold makes more sense biologically, no one would accept absolute threshold for population sparseness estimate. In any case, we have now done both and the results are actually comparable. We now include the new result in (Figure 3—figure supplement 1) for comparison. Showing both is important because they make clear that the sparse strong responses are meaningful for decoding both in relative and in absolute sense. Absolute threshold might be even more important the relative threshold for the biological reasons we mentioned earlier: downstream neurons might mostly care about the strength of the strong and robust input signals, and might not remember the “peak response” of each of its sources.

Furthermore, the discussion of the decoding results should be improved. Currently, it seems to imply a rather arbitrary threshold of around 20% that is considered 'good decoding' (e.g., in the comparison of the decoding results from the top 0.5% – which are around 20-30%, and the decoding results from the bottom 0.5% – which are around 15%). Both are far from the chance level, so these statements need to be further justified.

Our intent was to show that the top 0.5% of the responses alone can achieve 50% of the achievable decoding performance (28%/54% for monkey A, 21%/38% for monkey B), and that removing the top 0.5% of the response drop the accuracy to less than 50% of the achievable performance (19/54 for monkey A and 10/38 for monkey B). We believe the reviewer might have misinterpreted the blue curve, at 0.5% (where it intersects with the gray line), the blue curve means the accuracy when the top 0.5% response is removed, *not* the accuracy when the bottom 0.5% response is used. These two curves are used to demonstrate “sufficiency” (red curve), and “necessity” (blue curve) of the top 0.5%.

Finally, the authors conclude that the comparison of decoding performance for top and bottom responses demonstrates that strong responses are both necessary and sufficient to transfer relevant information. This is incorrect. The sufficiency is indeed demonstrated (accepting the assertion that decoding performance above some threshold constitutes successful decoding). However, to demonstrate necessity, they would have to demonstrate that successful decoding always implies the occurrence of strong responses. This is not the same as demonstrating that weak responses do not allow 'successful' decoding.

This issue might arise from the reviewer’s misinterpretation of the blue curve (see above). The blue curve indicates that without the top 0.5% of the strong response, decoding performance drops below 50% of the achievable decoding accuracy. At some level, this demonstrates the “necessity” of the top 0.5% for achieving good decoding, because without them the rest 99.5% of the population responses can realize only (or slightly less than) 50% of the achievable performance. We hope that with a better clarification of the blue curve, we can resolve the confusion, and that the reviewer will see the “necessity” of the strong response. In fact, the bottom 90% of the responses alone perform extremely badly (close to chance – <0.1%) (blue curve at 10% x-axis). On the other hand, in the presence of the strong response, the bottom 90% responses can contribute significantly, contributing over 20% of the achievable decoding performance (the red curve).

2d) It would be interesting to compare the sparseness of responses evoked by natural images to that evoked by gratings (which in other studies have been shown to drive a large percentage of superficial V1 neurons). This would also allow a better assessment of how many neurons could potentially respond, further alleviating concerns about cell health or other properties of the imaging region (although this concern is largely addressed by the fact that most neurons respond to at least some of the images in the natural image set).

Thanks for this suggestion. Actually, we have shown this result in our early paper (Neuron, 2017 and Current Biology, 2018). With a standard significance check (t-test or Anova), more than 90% cells did respond significantly to orientation gratings. But those responses were much weaker than the optimal strong responses elicited by natural images or complex shapes. That result could be logically and tightly connected with this study. But we are sorry that we could not add all those wonderful results into this paper anymore because of the limitation of publication policy.

2e) A famous paper by Quiroga, Reddy, Kreiman, Koch and Fried (2005) illustrated extremely high sparsity in cells that responded to images of Jennifer Aniston and Halle Berry. I'm actually surprised that there was no mention in the present paper of that finding by Quiroga. Although a direct comparison may not be appropriate given the differences in areas, it may still be informative to ask whether the V1 cells have greater sparseness than the Jennifer Aniston cell.

That is a very interesting point. In fact, we are aware of that paper, and cited the paper in the context of using similar ROC analysis to show that the sparse responses are not due to spurious chance occurrence. We also see the analogy as the reviewer, in the sense that both these MTC neurons and our V1 cells also exhibit similar high degree of stimulus specificity. But we feel it might be a bit too controversial to make a direct comparison. We now mention this potential connection in our Discussion/Conclusion paragraphs.

Another reason for connecting to the Quiroga paper is that they also do ROC analyses, but their ROC curves look very different than those of the present paper (see point 5e below). The comparison may provide further evidence that the sparseness calculations, including the ROC calculations, were done properly.

Please see the response in 5e below.

3) Clarification of experimental methods employed.3a) Cell count. The overall number of cells per imaging region is crucial for estimating sparseness. The ROI-definition procedure adopted by the authors appears reasonable and well justified. However, a few additional details would be useful:- Additional images of identified cells so that the accuracy of the chosen approach can be assessed.

We used the same algorithms to identify the neurons as reported in our previous work (Tang et al., 2017). All the cells with significant activities (over 3std) in differential images evoked by any stimuli could be identified automatically (ROIs). All the identified ROIs are shown in Figure 1C. We overlapped the ROIs to an image of the cells to show the ROIs extracted based on activities were well matched to cell bodies (Figure 1—figure supplement 3).

- How does the imaging region from monkey 2 look like?

We have added images from monkey 2 in new Figure 1—figure supplement 2.

- Which manual steps are involved in the procedure (presumably, somebody checks the identified ROIs)?

The ROI identification is automatic without manual operations.

- Were all data collected on the same day? If not, how are data and cell counts combined across days? How many cells were stable across those days?

Yes, all data were collected on the same day.

- How many of these cells are filled in versus have a ring-like label? This will help to assess how many of the included neurons are presumably healthy and should exhibit normal responses.

See Figure 1B, most of the cells (more the 95%) have ring-like label.

3b) Visual stimuli. Only the size of the stimuli are given. What are the other characteristics of the natural image set? Are they in color? Are they isoluminant with the background? What spatial frequencies and colors do they span? How different is their content? Are they part of one of the standard sets of natural images used in other studies?

Yes, they are in color (see the examples in Figure 1C, 1H and 1J). They are a large set of raw ‘natural images’. We did not modify them in color or contrast, as we tried to understand how the visual cortex works under natural conditions in this study. Those images are cropped from photos containing varying objects and scenes, with vary view distances, thus covered very large spatial frequencies and color ranges. Each stimulus is 4 x 4 degrees and hence 6-8 times larger than the receptive fields, which were placed at the center of the images.

The claim that single neurons respond to similar features of stimuli is not well supported and premature [“neuron 653 of monkey A was most excited when its receptive field (0.8o in diameter) covered the lower rim of the cat's 1 eye'. 'neuron 949 of Monkey A was found to be selective to an opposite curvature embedded in its preferred natural stimulus set”].

We have systematically investigated the pattern selectivity of V1 neurons in another study (monkey A of the two studies is actually the same, and many neurons overlap). We found most of the V1 superficial layer neurons exhibit high degree of selectivity to more complex and higher order patterns, such as corners, curvatures and radiating lines (Tang et al., 2018). Such selectivity could be the reason for the high degree of sparseness observed in this study. The two cells are just two typical examples for connecting the two studies. In the other study, each cell was tested with 9,500 patterns. The five patterns shown in Panel B and D of Figure 2 are just examples from the 9,500 tested patterns. More complete and detailed analysis of pattern selectivity of the neurons is presented in the other paper.

http://www.cell.com/current-biology/fulltext/S0960-9822(17)31521-X).

4) Points for further discussion/elaboration.4a) The Discussion is focused on issues of sparseness. However, there are two issues that are also worth mentioning. First, this study thus provides a fresh view of what V1 is doing, one that shifts the emphasis from simple orientation selectivity to complex natural stimuli, and which gives novel perspective on how the brain encodes natural stimuli. And second, the results show that, within the span of a single hypercolumn, all stimuli presented could be largely decoded. This supports Hubel and Wiesel's original concept that a hypercolumn contains all the machinery to encode everything at a single point in space, except that the manner of encoding may be distinct from (or more complex than) the original concept of selection from amongst an array of systematically organized ocular dominance, orientation, and color columns.

Thanks for these suggestions! We have made the first point in our Current Biology paper (Tang, et al., 2018) with explicit evidence on the pattern selectivity of the neurons. Second point is an interesting suggestion. While we completely agree with the reviewers’ interpretation, we were afraid the evidence presented in this paper by itself is not sufficient to make this statement.

4b) A very important item that should be made very clear is the last sentence of the Abstract where the authors correctly claim that this is the first paper that shows sparseness of neuronal population responses in V1 in primates. They need to point out that papers like Frouderakis et al. (2014) were in mice and papers like Quiroga, et al. (2005) were in humans but not in V1. The statement at the end of the tenth paragraph of the Results and Discussion needs to make that clear. It is such an important point that it needs to be pointed out in the Introduction and the Conclusion. Other researchers would then become aware of the power of two-photon imaging.

Thanks for these suggestions! Actually, we already mentioned this in the Introduction. But now we added the following as the first sentence in the “Conclusion”:

“In conclusion, while earlier studies provided life-time sparseness measurements in rodents (Hromadka et al. 2008; Haider et al. 2010), non-human primates (Rolls and Tovee, 1995; Vinje and Gallant, 2000; rust and DiCarlo 2012) and human (Quiroga et al. 2005), and population sparseness measurement in rodents (Froudarakis et al. 2014), our study provided the first direct measurement of sparseness of large-scale neuronal population responses in awake macaque monkeys, made possible by large-scale two-photon imaging.”

5) Clarification of data/analyses in the figures.5a) I find Figure 1F and G vs. L and M description very confusing. I think this is the interpretation: For population sparseness, one expects the distribution to show that for most images only few cells respond. For single cell selectivity, one expects each cell to respond to only a few images. Somehow the description of these graphs seems garbled. [E.g. then for each picture only 0.5% of cells responded means: in Monkey A, for 250 pictures 1 cell responded; for ~200 pictures 6 cells responded; for 5 pictures 20 cells responded. Alternative interpretation of graph: 1 cell responded to 250 pictures; 6 cells responded to ~200 pictures; 20 cells responded to 5 pictures?

The reviewer’s expectation is correct. “For population sparseness, one expects the distribution to show that for most images only few cells respond. For single cell selectivity, one expects each cell to respond to only a few images.” But the interpretation of the figure is wrong. We will try to make it clearer:

In such a frequency histogram graph, when there is a bar with height of 250 (Y-axis) at half-width of cells = 1 (x-axis), it means that there were 250 cases (pictures) with one picture activating only one cell strongly (among the 1225 cells in monkey A), i.e. 250 extremely sparse cases with only 1 cell responding to each picture.

We have also modified the captions of Figure 1G and G as follows: “(F and G) Frequency histograms showing the number of stimuli (out of 2250) (y-axis) that produce population responses with different population sparseness, measured in half-height bandwidth, i.e. the number of neurons activated strongly (x-axis). It shows that less than 0.5% of the cells (6 cells out of 1225 for monkey A, and 4.1 cells out of 982 for monkey B) responded above half of their peak responses for any given image on the average.”

I would not use 'half-height bandwidths'. Use 'number of cells'.

True. To minimize jargon, we changed half-height bandwidths to the percentage of the number of neurons or stimuli, which might be more explicit and easier to understand. We might still use half-height bandwidths at some places, because this is a standard expression in sparse code studies.

Why is stimulus specificity called 'life-time sparseness'? F-G, L-M should be described better to distinguish what each is saying (single cell selectivity vs. population sparseness/redundancy). Maybe partly it is the terminology that is used.]

Yes. We agree that life-time sparsity is a technical jargon in the field (Science 287: 1273- 1276, 2000; J Neurophysiol 105: 2907–2919, 2011). We equate it with stimulus specificity. However, the stimulus specificity might have the connotation that we know what stimulus feature the neuron is coding for. In sparse code study as those in Gallant’s or McCormick’s, the researchers often didn’t or couldn’t figure out the exact stimulus (which often include contextual surround) that give the neuron strong sparse response, hence they use the term life-time sparseness to indicate the neuron fire rarely to some specific stimulus, without claiming any knowledge of the neuron’s preferred stimulus.

5b) Figure 1: Why are there points below zero in D and E?

There are two possible reasons for the negative value in calcium imaging signal. First, some neurons may have spontaneous activities, and were depressed by some stimuli, which decreased the fluorescence signals (F < F0). Second, measurement noises could also make some signals negative.

5c) Figure 1: 'Cells 653 and 949 are colored red respectively.' Don't see this.

Sorry about the error. We have deleted this sentence.

5d) Figure 2: 0.5% contributes 50% of information and 5% contributes to 80% of information, what are remaining cells doing?

The responses of 0.5%-5% of the cells contribute to encoding of a given image. The responses of another 0.5%-5% of the cells contribute to encoding another image. Every cell is useful, just for different stimuli. In a way, this is very similar to the case of Quiroga’s paper where one cell was coding Jennifer Aniston, while another cell coding the Halle Berry, and yet another cell coding Sydney opera house, and these three cells did not care about other objects or concepts.

5e) Figure 1—figure supplement 2: Item C shows ROC curves for 99 shuffled trials. The part that wasn't at all clear to me was why did you compare the results to a shuffled version of the results.

The logic is, if the strong responses (say > half maximum for a neuron) in the raw data are random and spurious signals, then the curve of raw data will be comparable with those of shuffled version, and they will have similar ROC and AUC. On the other hand, if the strong responses for specific stimuli in raw data are reliable and repeatable across trials, the AUC will be significantly higher than those from shuffled data. The example shown in Figure 1—figure supplement 2, strongly against the hypothesis that the strong responses (three red data points in Figure 1—figure supplement 2A) came from random or spurious single trial epileptic responses.

And why did the shuffled data have such a high hit rate at zero false alarms. I would have thought that the shuffling would greatly reduce the hit rate. That is quite different from Quiroga's (2005) paper, which shows more normal curves with the false positive rate close to zero. If the authors are unable to use the Quiroga method then they should explain why, and why they end up with the very unusual shape for the ROC curves.

Yes, there are some key differences between Quiroga’s ROC analysis and ours. In Quiroga’s case, images from a certain individual or object formed the positive class, and images from others formed the negative; in our case, images with high enough (higher than half-maximum) mean responses form the positive class, and images with lower mean responses form the negative.

In Quiroga’s case, the set of positive images was *fixed* across all shuffles, and the authors tested whether the classification performance of a neuron for this *fixed* set of images against other images was by chance or not; in our case, the set of positive images *varied* across all shuffles, and we tested whether the classification performance of a neuron for high-*mean*-response images against others using *per-trial* responses was by chance or not. We cannot use Quiroga’s method for assigning positive and negative classes because our stimuli do not come from shared individuals or objects; because we changed the set of positive images for each shuffle, the ROC curves are biased to have high hit rates. Regardless of all differences, our ROC analysis is still valid, showing that the strong responses were *not* spurious or epileptic; otherwise, the shuffle operation would not change the AUC much.

To illustrate the point, let us imagine the following simplest case. Suppose a cell responding strongly to only one particular stimulus among 2,250, in every repeat and there are two repeats. There is then only one strong peak in the mean stimulus-tuning curve. The ROC curve will starts at (0, 1), with AUC (area under the curve) = 1. The shuffled version of the two trials will have two peaks (weaker but still significantly stronger than the rest) in the mean tuning curve. In this case, the ROC curve will start at (0, 0.5) with AUC near 0.75, as the hit rate is only 0.5 before any false positive appearing. When there are three repeats with one repeatable strong peak, the mean tuning curve of the shuffled case will have three peaks, as in our case, then the ROC will start at (0, 0.33) with AUCs around 0.667. This is because, as described above, the positive set of images is re-defined for the shuffled case (indicated by the blue spikes in Figure 2—figure supplement 1A), whereas the original positive set is indicated by the red spikes in Figure 2—figure supplement 1. In the real situation, as in the example shown in Figure 2—figure supplement 1, there are very strong peaks but also peaks close to the 0.5 max, the shuffled trials might contain some additional peaks due to noises or other low responses, making the ROC curve starting at (0, 0.33 – 0.4 range) and the AUC = 0.68 on average. When we increase the number of repeats, the starting point for the ROC will move toward (0, 0). In Quiroga’s case, the set of positive images was fixed, hence the ROC curves of shuffled responses start at (0, 0). That is the reason for the difference. In any case, the AUC of repeatable responses (across trials) will have significant higher AUC than that from shuffled trials, whereas spurious responses (unrepeatable across trials) will have similar AUC to that of the shuffled trials.

5f) Perhaps it would be helpful to the reader to have Figure 1 broken up into multiple figures. By doing that the figures would be larger with details more visible. One improvement that would be helpful for Figures1D, E would be to have the x-axis be a log axis so that the cell rank of the first 20 or so neurons would be more visible. I believe this is the plot that best demonstrates sparseness so it needs to be very clear.

Thanks for this constructive suggestion! We have separated Figure 1 into two figures. As for the suggestion of using log X for the neuronal response distribution plots (Figure 1D and E), as shown here, whilst using the log X allows easier visualization of the data, we feel it is not commonly used in the field for tuning curve, and might confuse some readers. We prefer to use the linear scale in the main text as it better convey the dramatic sharpness of the tuning curves and simplify the comparisons across studies.

**Author response image 2. respfig2:** The distribution of neuronal population responses with log X-axis.

I would suggest also spending substantially more effort clarifying F and F0 on which panels D and E are based. Another question is what is the role of noise for the one second intervals where the stimulus is shown? There is expected to be more noise in that interval than in the interval without the stimulus. How is that noise estimated?

We used a standard definition of F0, F and dF in this field. For each ROI, the F0 is the average fluorescence intensity during interval without the stimulus and F is average fluorescence intensity during interval with the stimulus. Yes, as shown in Figure 1I and K, the noises or variability of the signals in the interval with stimulus are larger than those in the interval without stimulus. That is consistent with situations in traditional electrophysiological studies.

5g) In connection with Figure 1—figure supplement 2 it would be useful to show A and B as histograms in addition to the way they are presently shown.

We have added histograms for top 3 responses for Figure 1—figure supplement 2A and B.

5h) I suspect Figure 2 is an important figure. However it wasn't at all clear to me how it was calculated. I did figure out that the small inset was a continuation of the large plot that stopped at 10%. I wonder of the inset could be removed if log axes were used for the x-axis and the data would go from say 0.1% to 100%. But more important is to clarify how the red and blue data were calculated.

We thank the reviewer for the advice. We have now improved the description in the main text. We thought about putting back the insets into the larger figures by using log axes; however, it was easier said than done as we found that to be messier overall.